# Maternal Effects and Trophodynamics Drive Interannual Larval Growth Variability of Atlantic Bluefin Tuna (*Thunnus thynnus*) from the Gulf of Mexico

**DOI:** 10.3390/ani14091319

**Published:** 2024-04-28

**Authors:** José M. Quintanilla, Ricardo Borrego-Santos, Estrella Malca, Rasmus Swalethorp, Michael R. Landry, Trika Gerard, John Lamkin, Alberto García, Raúl Laiz-Carrión

**Affiliations:** 1Instituto Español de Oceanografía (IEO-CSIC), Centro Oceanográfico de Málaga, 29640 Fuengirola, Spain; ricardo.borrego@ieo.csic.es (R.B.-S.); garaltomar@gmail.com (A.G.); raul.laiz@ieo.csic.es (R.L.-C.); 2Departamento de Biología Animal, Facultad de Ciencias, Universidad de Málaga, 29071 Málaga, Spain; 3Cooperative Institute for Marine and Atmospheric Studies, University of Miami, Miami, FL 33149, USA; estrella.malca@noaa.gov; 4NOAA Fisheries, Southeast Fisheries Science Center, Miami, FL 33149, USA; trika.gerard@noaa.gov (T.G.); j.lamkin@me.com (J.L.); 5Scripps Institution of Oceanography, University of California San Diego, La Jolla, CA 92037, USA; rswalethorp@ucsd.edu (R.S.); mlandry@ucsd.edu (M.R.L.)

**Keywords:** Atlantic bluefin tuna, larval growth, trophic ecology, isotopic signatures, maternal effects, trophic niches

## Abstract

**Simple Summary:**

Environmental factors, maternal inheritance, and feeding success are influential factors in fish growth, especially during the larval stage, encompassing their early days of life. Growth rates play a crucial role in larval survival, particularly in species with high energy requirements such as the Atlantic bluefin tuna (ABFT). Our analyses of two patches of ABFT larvae collected in the Gulf of Mexico’s spawning region during different years reveal variable larval growth, depending on prey availability. Larval growth also shows a direct relationship to maternal feeding. Estimates of larval trophic positions are primarily influenced by food web length and energy transmission efficiency, leading to differences in larval growth and underscoring the importance of considering trophic dynamics in interpreting results. These findings offer novel insights into how these factors affect ABFT larval growth, potentially informing conservation efforts and fisheries management strategies by governmental institutions.

**Abstract:**

Two cohorts of Atlantic bluefin tuna (*Thunnus thynnus*) larvae were sampled in 2017 and 2018 during the peak of spawning in the Gulf of Mexico (GOM). We examined environmental variables, daily growth, otolith biometry and stable isotopes and found that the GOM18 cohort grew at faster rates, with larger and wider otoliths. Inter and intra-population analyses (deficient vs. optimal growth groups) were carried out for pre- and post-flexion developmental stages to determine maternal and trophodynamic influences on larval growth variability based on larval isotopic signatures, trophic niche sizes and their overlaps. For the pre-flexion stages in both years, the optimal growth groups had significantly lower δ^15^N, implying a direct relationship between growth potential and maternal inheritance. Optimal growth groups and stages for both years showed lower C:N ratios, reflecting a greater energy investment in growth. The results of this study illustrate the interannual transgenerational trophic plasticity of a spawning stock and its linkages to growth potential of their offsprings in the GOM.

## 1. Introduction

Top predatory fishes in mid- to high-latitude marine ecosystems play crucial roles in the stability of food web structure [1,2]. For Atlantic Bluefin Tuna (ABFT, *Thunnus thynnus*), cascading effects of population fluctuations can alter the structure and performance of the lower food web [3,4,5].

ABFT is managed by the International Commission for the Conservation of Atlantic tunas (ICCAT) as two, eastern and western, stocks [6] with different natal homing behaviors, spawning areas, sexual maturity ages, and trophic dynamics [7,8,9]. The western stock feeds principally in prey-rich waters of the north and northwestern Atlantic [10,11,12,13]. Displaying a capital feeding strategy for reproduction [14,15], nutritionally replenished adults migrate thousands of kilometers each year to reproduce in warm oligotrophic waters of the Gulf of Mexico (GOM) [16,17].

While nutrient poor [18,19], oceanic waters in the GOM spawning grounds benefit from proximity to richer shelf region with inflows from numerous rivers like the Mississippi [20,21,22,23,24,25] and by the direct influence of the Loop Current, which forms eddies and gyres and shapes mesoscale hydrographical circulation [26,27,28]. These mesoscale features serve as ideal nursery habitats for bluefin tuna larvae [29], with spawning occurring from April to June [30] when surface temperatures exceed 24 °C [17].

The two main environmental influences on the early life stages of tunas are temperature and food availability [31,32,33,34,35]. Temperature influences vital and metabolic rates that in turn affect rates of growth and mortality [36,37]. Temperature enhances growth rate of tuna larvae when food availably is sufficient [31,33,38] and is the main abiotic driver of tuna distribution and recruitment [39]. Successful and frequent feeding during early life history depends on adequate food resources and is essential for survival.

Knowledge of the relationship between growth and trophic ecology is fundamental for understanding how larvae respond to varying spatio-temporal dynamics of their nursery habitat [40,41]. Environmental impacts on the stock–recruitment relationship for ABFT result in varying recruitment scenarios. Understanding larval survival rates and the stock-recruitment relationship in their spawning grounds informs management decisions on fishing pressure and stock recovery potential and is critical for effective management.

Trophic studies in fish larvae have mostly focused on stomach content analyses which give snapshots of prey consumed over relatively short feeding periods [42,43,44,45]. Stable isotope analyses (SIA) complement traditional gut content examination with biogeochemical information on the mean trophic characteristics of prey consumed over a longer time scale, essentially the nutritional history of the larvae up to the point of capture [14,46,47].

Nitrogen (δ^15^N) and carbon (δ^13^C) SIA are often used to assess trophic position and carbon flows to consumers in food webs [48,49,50]. Nitrogen δ^15^N is an indicator of the mean N sources supporting consumer growth and enriches with each trophic transfer in the food chain. Since C isotope ratios undergo small changes during trophic transfers, δ^13^C is mainly used to assess how food sources with different mean δ^13^C values contribute to diet [49,51]. SIA has been previously applied to evaluate trophic influences on larval growth of ABFT using size-fractionated zooplankton as baseline values [52].

Isotopic signatures (δ^15^N and δ^13^C) evolve as different prey types are selected by larval of increasing size and developmental stage [53,54]. For pre-flexion stages, δ^15^N signatures are derived principally from maternal transmission [55], corresponding to the consumer isotopic signatures of adult breeding females. In contrast, δ^15^N for post-flexion stages, increases with size and development and reflects larval dietary changes [53]. In field samples, this trophic change is interpreted as a tendency toward larger prey-size consumption with age [54,56]. These larval δ^15^N values, together with the baseline isotopic signature of the micro-zooplankton community (0.05–0.2 mm in size), allow us to estimate and compare TPs among populations and to understand the ecological roles of different species in the system [50,57], their trophic structure and consumer–prey relationships [58]. Therefore, TP estimation is crucial for understanding trophodynamics and the influence of trophic interactions on larval growth variability.

Stable isotopes are also used to estimate trophic parameters, such as maternal/trophic isotopic niche widths and niche overlap, by applying stable isotope Bayesian inference analysis [59]. These niches are measures of dietary diversity [60,61,62] and describe the isotopic characteristics of the niches exploited by the breeders (maternal) and larvae (trophic) of the species.

In this study, growth variability of two ABFT larval cohorts (2017 and 2018) are compared from two complementary perspectives: inter and intra-population analysis. We analyzed larval daily growth with trophodynamics characterized by SIA analysis and isotopic niches considering both the total population (TOTAL) and the segregated groups of pre-flexion (PRE) and post-flexion (POST) larvae, which reflect maternal and larval trophodynamic influences, respectively.

## 2. Materials and Methods

### 2.1. Sampling and Processing of ABFT Larvae and Plankton

Samples of ABFT larvae and zooplankton prey were collected in the GOM spawning region at 9 stations on BLOOFINZ cruise NF1704 (May 2017) and at 19 stations on BLOOFINZ cruise NF1802 (May 2018) aboard NOAA R/V Nancy Foster (Figure 1). On each cruise, we first located a patch of significant larval abundance with preliminary net tows, then marked the patch with a free-floating satellite-tracked drifter with a 3 m drogue centered at 15 m in the surface mixed layer and repeatedly sampled the larvae and ambient zooplankton prey in the same water over the course of 3–4 days [63]. Larval patches GOM17 and GOM18 in Figure 1 were found at different locations; GOM18 being closer to the richer continental margin than GOM17, which was well into oligotrophic waters of the central GOM. However, both larval patches were found to have originated ~2–4 weeks earlier at roughly the same location along the northwest continental margin of the GOM by backtracking drift trajectories in a reanalysis of surface water circulation [63]. This study therefore compares two groups of ABFT larvae originating from the same general location in different years but experiencing different trophic conditions.

For ABFT larvae, we used a Bongo 90 cm net frame with 500 μm mesh towed obliquely from the surface to 25 m and back at approximately 2 knots for 10 min. Zooplankton samples were collected on the same tows as the larvae, using a 20 cm Bongo net frame with 200 and 55 μm mesh nets attached to the Bongo 90 cm net frame [52]. Each of the Bongo 90 and Bongo 20 nets were equipped with a General Oceanics flowmeter to measure the volume of water filtered during each tow (m^3^). Temperature (°C) and salinity (psu) profiles for the upper 25 m were determined from CTD casts conducted concurrently at each station (see [64] for hydrographic sampling details).

ABFT larvae were sorted, preserved and processed on board following standard protocols [65] to obtain standard length (SL, mm) and dry weights (DW, mg). Specimens were freeze-dried and placed in individual tin capsules (0.2–2 mg) for SIA analyses.

Zooplankton from each net were spilt in two subsamples [45]. From the 55 μm mesh, one subsample was frozen at −20 °C for biomass and SIA, and the other preserved with 4% formaldehyde for community analysis. The two subsamples from the 200 μm mesh net were concentrated and frozen at −20 °C and preserved in ethanol 96%, respectively.

### 2.2. Otolith Analyses

Otoliths were removed, cleaned with distilled water and fixed on slides with one drop of nail lacquer [65]. Sagittal otoliths were digitalized as stacks of focal-depth images, varying in number depending on otolith size. Otoliths were excluded if they were broken, not saggitae or had fixation artifacts. Otolith radius (RADIUS, µm), daily increments (AGE, days) and mean increment widths (MIW, µm) were also measured by Leica image analysis software. Reading criteria for ABFT larval age estimations were previously applied [34,65,66] and detailed by Malca et al. [64].

### 2.3. SIA Analyses of Larvae and Zooplankton

Natural abundance of N (δ^15^N) and C (δ^13^C) were measured with an isotope-ratio spectrometer (Thermo-Finningan Deltaplus) coupled to an elemental analyzer (FlashEA1112 Thermo-Finningan) at the Instrumental Unit of Analysis of the University of A Coruña. Ratios of ^15^N:^14^N and ^12^C:^13^C are expressed in conventional delta notation (δ), relative to the international standard [atmospheric air (N_2_) and Pee-Dee Belemnite (PDB), respectively, using acetanilide as standard]. The analytical precision for δ^15^N and δ^13^C were 0.13 and 0.11‰, respectively, based on the standard deviation of internal references (repeatability of duplicates). A posteriori corrections of δ^13^C values for lipid content were carried out based on C:N ratios for micro- and meso-zooplankton size fractions according to the equations and parameters for invertebrates [67] and for muscle tissue of ABFT larvae [52].

### 2.4. Estimation of Isotopic Maternal Signatures

We estimated isotopic maternal values using the model of Uriarte et al. [53]:δ^15^N_maternal_ = δ^15^N_larvae_ + (δ^15^N_egg_ − δ^15^N_larvae_)
δ^13^C_maternal_ = δ^13^C_larvae_ + (δ^13^C_egg_ − δ^13^C_larvae_)
where δ^15^N_larvae_ and δ^13^C_larvae_ are bulk SIA values for each larva. To calculate the factors (δ^15^N_egg_ − δ^15^N_larvae_) and (δ^13^C_egg_ − δ^13^C_larvae_) required for estimating maternal isotopic values, we used the isotopic values for each pre-flexion larva on each survey to determine a linear relationship of isotope variability with age (Table 1).

The δ^15^N and δ^13^C values for eggs were obtained from newly spawned eggs and lecithotrophic larvae (n = 20 pooled) from aquaculture rearing experiments [53]. For wild ABFT larvae, the isotopic values of eggs were calculated using a random variable originating from the mean and standard deviation of egg and lecithotrophic larvae of the rearing experiment [65].

### 2.5. Larval Trophic Positions

The trophic position (TP) of each ABFT larvae was calculated following Equation (1):TP = ((δ^15^N_larvae_ − δ^15^N_micro_)/Δ^15^N) + TP_basal_(1)
where δ^15^N_larvae_ is the larval N isotopic signature and δ^15^N_micro_ is the isotopic value for micro-zooplankton at the same station. We applied a basal trophic position (TP_BASAL_) of 2, assuming micro-zooplankton are primary consumers [68]. For the nitrogen isotopic discrimination factor (Δ^15^N), we used the muscle tissue value for juveniles ABFT (1.46‰) proposed by [69] and previously applied to ABFT larvae by [64].

### 2.6. Maternal and Larval Isotopic Niche Widths and Overlaps

Maternal isotopic niches widths were estimated from δ^15^N_maternal_ and δ^13^C_maternal_ values, calculated from the isotopic values of pre-flexion larvae. Larval isotopic niche widths were calculated from δ^15^N and δ^13^C values of post-flexion specimens’ stages to avoid the maternal influence. The isotopic niche widths were estimated by standard Bayesian ellipse areas and associated credible intervals were adjusted for small sample size (SEAc) [59,67]. Isotopic niche widths and overlap analyses were conducted using the R package SIBER (Stable Isotope Bayesian Ellipses in R) v.3.3.0 ([59], R Development Core Team 2012). Standard ellipses were calculated from the variance and covariance of 40% of the bivariate data.

### 2.7. Statistical Analyses

Environmental variables were compared with the non-parametric Mann–Whitney U test as the variables did not meet parametric assumptions. Significance tests for the differences’ growth, isotopic signatures, C:N values and otolith metrics between GOM17 and GOM18 larval groups were performed by ANCOVA using AGE as the covariable. The variables were Log transformed prior to statistical analyses when it was necessary to obtain linearity and variance homogeneity [70]. When there was no linear relationship of a variable with AGE, an ANOVA analysis was used to determine differences between groups.

We used linear equations (y = a + x*b) for LogSL and LogDW vs. AGE to define the daily growth pattern of each larval cohort, and their residual values were obtained with respect to the whole population. GOM17 and GOM18 cohorts were divided into four groups according to their residual values of length (SL) and weight (DW) controlled by AGE. Larger and heavier than expected larvae were in the OPT group, with positive residuals for both fits, while smaller and lighter than expected larvae were in the DEF group, with negative residuals for both fits. Two intermediate groups (shorter SL but heavier and vice versa) were not considered in this study. Following the method of [71], the residual analysis defined groups according to the optimal (OPT) and deficient (DEF) growth patterns in length (SL) and weight (DW) of individual larvae. For intra-population comparisons, these groups were compared through an ANCOVA analysis controlled by AGE [65].

For isotopic niche widths, the color scale (dark, medium and light) represents confidence intervals of 50%, 75% and 95%, respectively. Isotopic niche widths and overlap analyses were conducted using the R package SIBER (Stable Isotope Bayesian Ellipses in R) v.3.3.0 ([59], R Development Core Team 2012).

Statistical analyses were carried out using R version 4.2.1 (R Core Team 23-6-22 ucrt) through the integrated development environment RStudio, with α = 0.05.

## 3. Results

### 3.1. Environmental and Abiotic Variables

Environmental variables showed significant differences between years with higher temperatures and lower salinities for GOM18. The isotopic signatures of micro and meso-zooplankton fractions were higher in δ^15^N and lower in δ^13^C for GOM18 (Table 2).

### 3.2. Larval Growth

Larval growth showed a normal distribution with a common size range (4–9 mm) for both groups (Figure A1). For the TOTAL group, somatic and otolith biometrics differed between years, with higher values of SL, DW, RADIUS and MIW in GOM18 (Figure A2 and Table 3).

For pre-flexion larvae, the SL did not differ between the years, while GOM18 had higher DW and GOM17 had larger RADIUS and MIW (Figure A3 and Table 3). In contrast, somatic and otolith variables were both consistently higher in GOM18 for post-flexion (Figure A4 and Table 3). At the intra-population level, every growth pattern differed for both development stages, showing higher mean values for OPT larvae (Figure A3 and Figure A4 and Table 4).

### 3.3. Larval Trophic Variables

The inter-population δ^15^N and δ^13^C values were higher for GOM18 for both pre- and post-flexion stages and TOTAL larvae (Figure 2 and Table A1). Despite GOM18 larvae having higher δ^15^N (6.60 ± 0.78 vs. 4.47 ± 0.60), TP values were higher for GOM17 (4.12 ± 0.25 vs. 3.47 ± 0.22) (Figure 2 and Table A1). At the intra-population level for both GOM17 and GOM18, DEF larvae had higher levels of δ^15^N and C:N, while δ^13^C levels were higher for OPT (Figure 2 and Table A2). We found no intra-population TP differences for either year (Figure 2 and Table A2).

For GOM17 pre-flexion larvae, the DEF group had higher values of δ^15^N, δ^13^C and C:N than OPT. Values of δ^15^N were also higher for the DEF group in GOM18, but δ^13^C and C:N were not significantly different (Figure 2 and Table A2).

For GOM17 post-flexion larvae, C:N was higher for DEF, but no significant differences were found for δ^15^N, δ^13^C or TP. For GOM18 post-flexion larvae, δ^13^C and C:N were higher for the DEF group while δ^15^N and TP did not differ between DEF and OPT (Figure 2 and Table A2).

### 3.4. Trophic Niches

#### 3.4.1. Maternal Trophic Niches

For inter-population comparisons, maternal isotopic signatures showed ellipse area overlaps of 60% (0.49) (Figure 3A) and no significant differences in δ^15^N and δ^13^C values between the years (Table 5). The estimated niche area for GOM17 was slightly larger (0.67) than GOM18 (0.65) (Figure 3B).

Comparing optimal growth (OPT) and deficient (DEF) groups, estimated maternal values of δ^15^N were significantly different between the years, with higher values in the DEF groups. However, estimated maternal δ^13^C values were not significantly different between OPT and DEF for both years (Table 6).

For GOM2017, maternal niches of the contrasting growth groups overlapped 14% (0.15) (Figure 4A), and were larger for DEF (0.89) than OPT (0.31) (Figure 4B). For GOM18, OPT and DEF larvae showed maternal niches of similar size (0.38 OPT vs. 0.36 DEF) (Figure 4A) with no overlap between them (Figure 4B).

#### 3.4.2. Larval Trophic Niches

Comparing inter-population trophic niches, we observed significant differences in both δ^15^N and δ^13^C between years (Table A1), without overlap of niche ellipses (Figure 5A). GOM18 larvae had larger trophic niches (0.58) than GOM17 larvae (0.23) (Figure 5B). At the intra-population level, OPT and DEF of both years did not differ in δ^15^N values (Table A2). In contrast, δ^13^C values differed between OPT and DEF for GOM18 (Figure 6A and Table A2). The trophic niches of OPT and DEF larvae for GOM17 overlap 36% (0.13) (Figure 6A), with similar niche sizes for both groups (0.25 OPT vs. 0.23 DEF) (Figure 6B). OPT and DEF larvae for GOM18 have a niche overlap of 19% (0.13) (Figure 6A), with larger areas for DEF (0.53) than OPT (0.28) (Figure 6B).

## 4. Discussion

Larval fish growth during early development is closely linked to their survival [72]. With increasing size, larvae become more adept at escaping predators and at catching larger, more nutritious prey and ward off starvation [73,74]. Thus, the faster the larvae grow, shortening the duration of this critical period of their lives, the lower the cumulative mortality during the larval stage [75,76].

Larval growth is characterized by great plasticity, which is reflected in its variability depending on environmental characteristics [77], with temperature and food availability being the most decisive factors for larval growth [31,32,33,34,78]. Temperature is an important factor for larval survival of the genus *Thunnus* sp. [38,79] and especially influential for early developmental of ABFT [80,81]. GOM18 larvae experienced warmer temperature (Table 2) and showed higher somatic growth (SL and DW) and larger otolith biometrics (RAD and MIW) (Table 3). However, the temperature differences between GOM17 and GOM18 were relatively small, only 0.84 °C on average, so it is unlikely to be the main reason for the differences in observed the growth patterns [56,82]. In previous studies of the same species, no growth pattern differences were detected between years with temperature differences exceeding 1 °C [33]. 

It is more likely that the inter-annual differences in growth rates would be better explained by differences in trophic dynamics [82,83] or genetic factors associated with maternal inheritance [65].

Concentrations of meso-zooplankton (0.2–1 mm) were higher for GOM18 [66] than GOM17, which could suggest a cause–effect relationship between potential food availability and growth variability. Field studies indicated that environmental factors account for less than 40% of the variability observed in larval growth [84,85], which suggests the importance of other factors such as genetic heritance. Maternal stable isotope transmission has been traced in perciforms to offspring [55]; however, few studies have applied stable isotopes to investigate the effects of maternal nutrition on offspring quality [86]. Uriarte et al. [53] showed in a rearing experiment that eggs and pre-flexion larvae of eastern ABFT larvae reflected the adult female isotopic signatures. For ABFT, maternal influence analyzed from the changes in N isotopic values during the pre-flexion stages has a decisive importance for larval growth [65]. This maternal effect gradually disappears with development until the values of the N and C isotope reach a steady state with exogenous feeding in post-flexion larvae. Therefore, to analyze the factors that determine the ABFT larval growth, it is useful to consider the pre- (greater maternal influence) and post-flexion (greater trophic influence) stages separately [14,53,54].

### 4.1. Maternal Influence (Pre-Flexion Larvae)

The maternal effect influences size at hatch and subsequent growth [87], thereby increasing larval viability and decreasing mortality [36,55,88]. The decreasing values of δ^15^N with pre-flexion age, together with an increasing δ^13^C profile for both GOM17 and GOM18 cohorts, agree with observations for the same species both in culture experiments [53] and in field studies [14,54,65]. The estimated maternal isotopic signatures (δ^15^N_maternal_, δ^13^C_maternal_) based on the equations of isotopic values with age in our field-collected samples (Table 1) showed no differences between years (Table 5), and the values were comparable to the isotopic signatures previously reported for adult muscle tissue [81,89,90,91,92,93].

Maternal trophic niches express the isotopic characteristics of the trophic niches exploited by the breeders, and their sizes can vary depending on food availability [94]. According to our results, the maternal isotopic niche areas were similar between years (Figure 3B) with a high degree of overlap (Figure 3A), which we interpret as the females feeding on prey with similar mean isotopic characteristics. Our results do not mean that the spawning adults in both years came from exactly the same geographic locations, but they do support the “common feeding grounds” hypothesis [16] by which adults aggregate in large groups to feed broadly in the western Atlantic Ocean [8,95,96].

Therefore, at the population level, it does not seem that growth differences for pre-flexion stages (Figure A3 and Table 3) were related to differences in maternal isotopic niches or to breeder trophic behavior. However, a direct relationship between maternal inheritance and larval growth can be evaluated by considering the contrasting growth groups in residual analysis [65,71].

According to our intra-population comparison, OPT growing larvae showed lower estimated maternal δ^15^N values in both years (Table 4), implying a direct relationship between growth potential and maternal inheritance. This variability in maternally inherited δ^15^N values may be based on various factors such as differences in age, condition, or natural variability in quality of spawning episodes [65].

In the GOM, ABFT is an opportunistic and generalist predator [91] that feeds on a wide range of available prey, and its diet is affected by food availability. The smaller maternal isotopic niches estimated for GOM2017 OPT larvae (Figure 4A,B), could be associated with a more selective maternal diet on a low number of species [97]. On the other hand, the wider maternal isotopic niches for DEF suggest a more diverse diet and generalist trophic behavior. As trophic niche size is related to ecosystem productivity [98], consumers must adapt a foraging strategy in order to satisfy their metabolic demands. Following this reasoning, larvae with greater growth potential would seem to be associated with more stenophagous maternal trophic behavior in which females cover their energy requirements with more selective feeding behavior in areas of greater production and likely higher prey quality. In contrast, those with lower growth potential would be associated with maternal euryphagous behaviors in which females search for food over larger less-productive areas.

Since 2018 OPT and DEF larvae had similar maternal niches widths, growth differences between these groups cannot be associated with differences in the breeder trophic behavior as for GOM17 (Figure 4A,B). Moreover, the absence of overlap of maternal niches between these groups (Figure 4A,B) could be due to many factors that determine N isotopic signature such as age, nutritional status and quality variance among/within spawning batches, as previously mentioned. Further investigations are needed to elucidate the implications of these various factors for larval growth variability.

### 4.2. Trophic Behavior (Post-Flexion Larvae)

δ^15^N levels are enriched with each trophic transfer, providing information about the TPs of consumers [50,99]. In previous studies, better larval growth was found to be associated with higher TPs [56,64,100], which was interpreted as reflecting greater trophic specialization. According to our results, however, larvae with better growth from GOM18 had lower TPs than GOM17 (Table A1). Similar observations have been reported for larval Shortbelly Rockfish where larvae with a lower TP were heavier and grew faster [101]. The TP reflects how the energy is transferred from the base of the food web up to the larva [99,102], and its estimation can be influenced by food chain efficiency [103,104], which causes a wide range of TP estimates in the GOM [64].

In the oligotrophic microbially dominated ecosystems in which ABFT larvae develop [16], the main trophic pathway is highly inefficient, with most production lost to bacterial remineralization and multi-step protistan food chains [105]. Knapp et al. [106] found that N_2_ fixation accounted for a relatively small component of new N-based productivity during GOM17 and GOM18, and Kelly et al. [107] observed that chronic N deficits in the offshore oligotrophic waters where ABFT larvae live are met by lateral advection of organic matter from the more productive shelf regions. The average contribution of nitrogen fixation was double in GOM17 compared to GOM18, while the advected particulate organic nitrogen (PON) was five times higher in GOM18 [104]. Thus, greater oligotrophy in 2017 could explain the unusually depleted values of δ^15^N observed from micro- and meso-zooplankton (Table 2) to ABFT larvae (Table A1).

Our results are consistent with those summarized by Gerard et al. [63] for the BLOOFINZ-GOM cruises, suggesting ABFT larvae are more likely to thrive by feeding at a lower trophic position regardless of the source of production (GOM18). These findings appear consistent with the newly proposed Trophic Efficiency in the Early Life hypothesis in which larval survival is linked to feeding on prey that are low in the food chain, thereby maximizing energy transfer [103] (Figure A2 and Table 3).

ABFT larvae are daylight visual feeders that feed selectively on preferred prey such as copepods, copepod nauplii, appendicularians and cladocerans [43,44,45,66,108,109]. Stukel et al. [104] indicated that an ABFT diet of calanoid copepods and podonid cladocerans (which were more abundant in the water column during GOM18), was consistent with maintaining relatively low trophic positions. Likewise, Shiroza et al. [45] demonstrated that highly selective predation on cladocera was an active process, which would support the idea that ABFT larvae are highly specialized for maximizing trophic efficiency in the oligotrophic environments where they develop.

In our inter-annual comparison, ABFT larvae occupy completely separate isotopic niche areas (Figure 5A), which would imply that they exploit prey with very different isotopic characteristics depending on the year. Zooplankton biomass was higher with greater diversity and concentrations of preferred ABFT larval prey for GOM18 compared to GOM17 [45,110]. The greater growth observed in 2018 would be associated with larger larval trophic niches that reflect greater availability of preferred prey, facilitating higher growth rates [66]. Conversely, the lower prey concentration and richness (including preferred types) in 2017 could result in narrower isotopic niches (Figure 5B) being, from a trophic point of view, a limiting situation for development, reflected in lower growth rates. In this case, inter-population growth differences in post-flexion stages would be associated with the aforementioned differences in food availability and diversity rather than with trophic behavioral shifts.

The intra-population analysis of isotopic niches offers different results for each group. In 2018, larvae with optimal growth are associated with narrower trophic niches, which can be interpreted as more selective trophic behavior in a higher production ecosystem (Figure 6A,B). At this point, it is important to highlight that the trophic niche differences between OPT and DEF are determined mainly by the range of variation in δ^13^C values, which reflect the prey carbon sources for larval growth. In this sense, a trophic niche size of OPT may rely more on food chains fueled by production from laterally transported water masses, which one might expect to have a higher δ^13^C signature [104,111].

On the other hand, the similarities of isotopic niches with respect to their widths and their high degree of overlap for GOM17 do not explain the observed larval growth differences between OPT and DEF groups for this year based on trophic criteria (Figure 6A,B). This is one area that could be better explained by maternal inheritance. This inter-generational transfer tracked through the δ^15^N values would arise from the trophic characteristics of breeders [65] that influence larval growth into the flexion stage. Moreover, it cannot be ruled out that the suboptimal larval feeding conditions and more homogeneous oligotrophic environment due to less lateral transport [104,110] prevent us from statistically separating OPT and DEF based on their isotopic niches. Additional research is required to clarify the effects of other factors on larval growth variability, such as population genetic variability, density dependent effects and early life thermal history.

C:N ratio has been used to evaluate nutritional status [112], being a particularly good proxy for the amount of lipid reserve [67]. For the intra-population comparison, C:N values were consistently higher in larvae with lower growth potential in both years and regardless of developmental stage (Figure 2 and Table A2). We interpret these results as lower consumption of lipid reserves for growth by DEF larvae, in contrast to the OPT growth group whose lipid levels are reduced as a consequence of a greater energy investment in somatic growth.

This study focused on answering open ABFT larval ecology questions crucial to understanding larval survival. There is an important lack of knowledge about the trophic implications for survival in the oligotrophic environments in which these larvae develop. We used larval growth and trophic analyses to describe growth variability with direct implications for larval survival and recruitment. Improved understanding of the connections among environmental variability, larval ecology and recruitment processes can inform future management strategies, including development of bluefin larval indices [113] that aid in meeting ICCAT sustainability goals within an integrated ecosystem-based approach.

## 5. Conclusions

Our results confirm a direct relationship between growth potential and δ^15^N signatures of ABFT pre-flexion larvae due to breeder feeding behavior that is transmitted by maternal inheritance. Maternal isotopic signatures and estimates of isotopic niche space of pre-flexion larvae are consistent with previous SIA studies of adult females.

Larval trophic ecology is reflected in isotopic niches widths and overlaps, which follow the availability and diversity of prey and their effects on larval growth potential. TP estimates are determined by food web length/efficiency and range substantially with temporal and spatial variability in trophic conditions. Regardless of developmental status, larvae with higher growth potential have significantly lower C:N, consistent with greater energy investment in growth.

## Figures and Tables

**Figure 1 animals-14-01319-f001:**
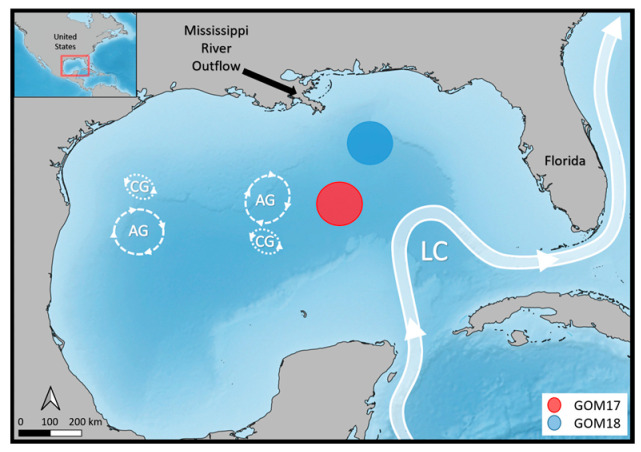
Area of larval ABFT tuna sampling stations during the BLOOFINZ surveys 2017 and 2018. Anticyclonic eddies (AG, dashed line) and cyclonic eddies (CG, dotted line) driven by extensions and contractions; the Loop Current (LC) is highlighting the hydrodynamic features of the GOM (Modified from [63]).

**Figure 2 animals-14-01319-f002:**
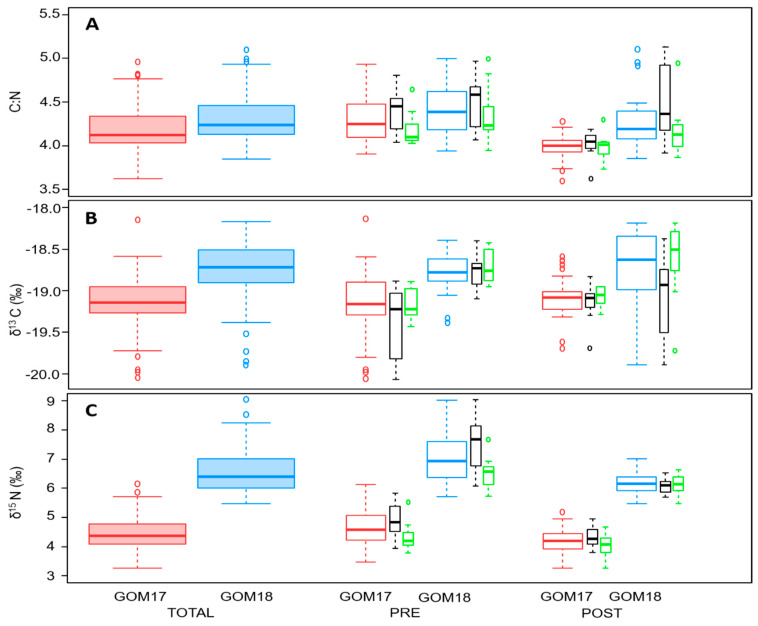
Boxplot of (**A**) C:N, (**B**) δ^13^C and (**C**) δ^15^N values of inter (GOM17red, GOM18—blue) and intra-populations (OPT—green, DEF—black).

**Figure 3 animals-14-01319-f003:**
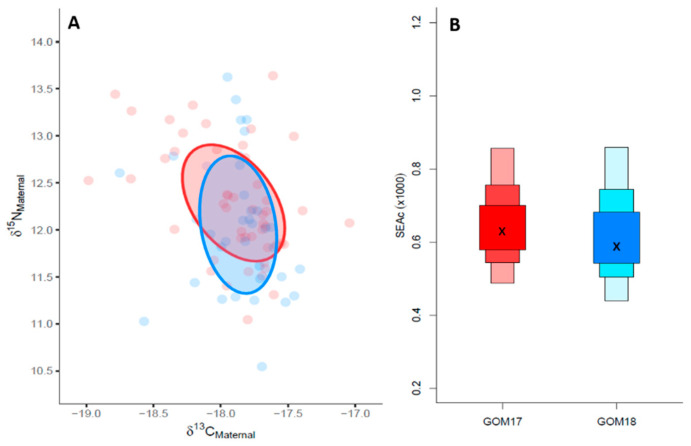
(**A**) δ^15^N vs. δ^13^C maternal values for GOM17 and GOM18. Maternal trophic niches are represented by the ellipse areas. (**B**) Estimated ellipse areas applying the correction for small sample sizes (SEAc).

**Figure 4 animals-14-01319-f004:**
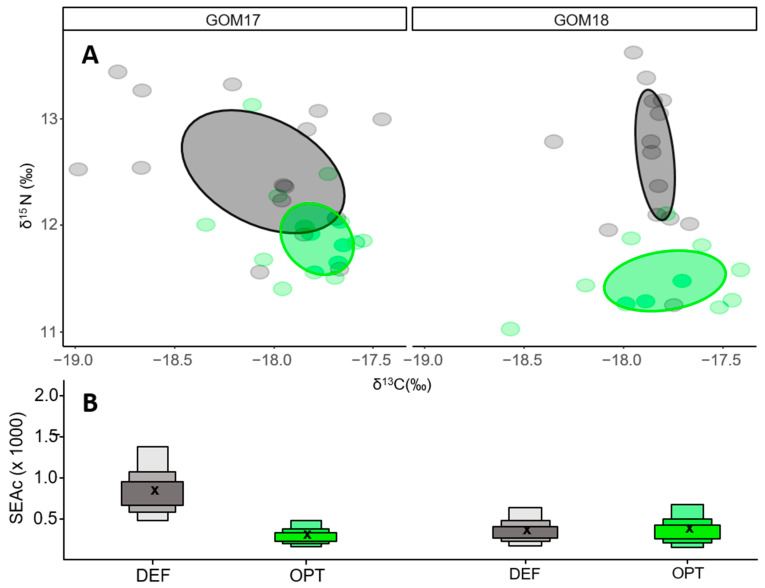
(**A**) δ^15^N vs. δ^13^C maternal values OPT (green) and DEF (grey) larvae for GOM17 and GOM18. The ellipses represent areas of maternal trophic niches. (**B**) Areas of the trophic niches estimated for OPT and DEF groups. The cross represents the size of the ellipse by applying the correction for small sample sizes (SEAc).

**Figure 5 animals-14-01319-f005:**
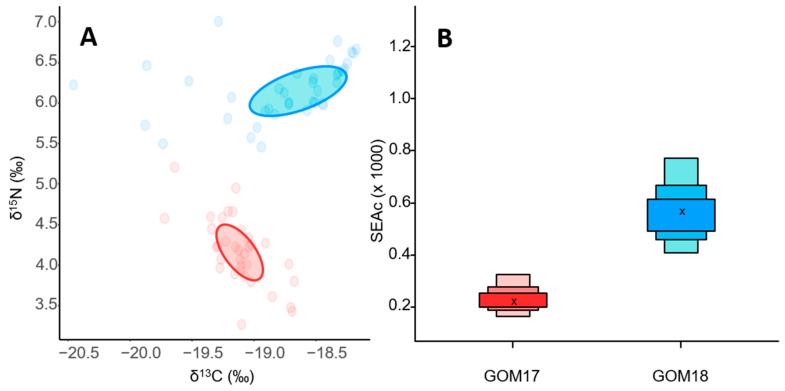
(**A**) δ^15^N vs. δ^13^C larval values for post-flexion larvae of GOM17 and GOM18. The ellipses represent the areas of the larval trophic niches estimated for each campaign. (**B**) Areas of the trophic niches estimated by SIBER of each GOM17 (0.23) and GOM18 (0.58) campaign. The cross represents the size of the ellipse by applying the correction for small sample sizes (SEAc).

**Figure 6 animals-14-01319-f006:**
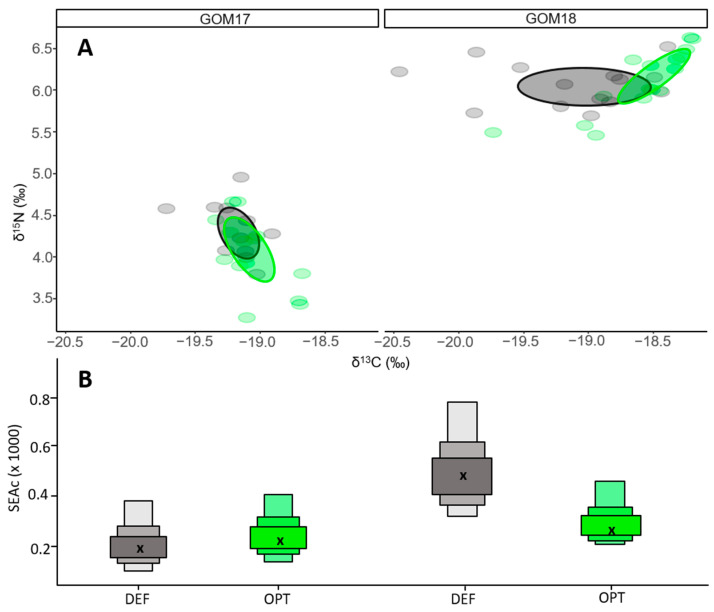
(**A**) δ^15^N vs. δ^13^C larval values OPT (green) and DEF (grey) larvae of GOM17 and GOM18 campaigns. Ellipse areas represent larval trophic niches. (**B**) Trophic niche areas estimated by SIBER with the correction for small sample sizes (SEAc).

**Table 1 animals-14-01319-t001:** Maternal isotopic signature equations derived from larvae captured in the field (GOM17 and GOM18). NS = Non-significant; * *p* < 0.05; ** *p* < 0.01.

Population	n	Maternal Isotopic Signature Estimation Equation	*p*	R^2^
GOM17	49	(δ^15^N_EGG_ − δ^15^N_LARVAE_) = (7.206 + 0.047 * AGE)	NS	0.01
		(δ^13^C_EGG_ − δ^13^C_LARVAE_) = (0.467 + 0.091 * AGE)	*	0.09
GOM18	52	(δ^15^N_EGG_ − δ^15^N_LARVAE_) = (0.975 + 0.527 * AGE)	**	0.52
		(δ^13^C_EGG_ − δ^13^C_LARVAE_) = (2.423 – 0.200 * AGE)	**	0.45

**Table 2 animals-14-01319-t002:** Mean values (Mean ± SD) of temperature (TEMP, °C), salinity (SAL, ppt), and δ^15^N and δ^13^C of micro-zooplankton and meso-zooplankton. U Mann–Whitney Test. ** *p* < 0.01.

	GOM17	GOM18	MW—U test	
	Mean ± SD	Mean ± SD	*Z-adjusted*	*p*
TEMP (C°)	24.69 ± 0.67	25.53 ± 0.47	−2.72	**
SAL (ppt)	36.38 ± 0.06	36.02 ± 0.32	3.30	**
δ^15^N_micro_	0.56 ± 0.42	3.61 ± 0.39	−3.00	**
δ^13^C_micro_	−18.1 ± 0.47	−19.3 ± 0.27	3.00	**
δ^15^N_meso_	1.75 ± 0.51	4.69 ± 0.42	−2.74	**
δ^13^C_meso_	−17.7 ± 0.52	−19.6 ± 0.25	2.74	**

**Table 3 animals-14-01319-t003:** Larval somatic data (SL, DW) and biometric otoliths (RADIUS, MIW) for GOM17 and GOM18 grouped by stage (PRE, POST, TOTAL). ANCOVA results (F, *p*) using AGE as covariate. NS = Non-significant; * *p* < 0.05; ** *p* < 0.01.

		GOM17				GOM18				ANCOVA	
		**n**	**Min**	**Max**	**Mean ± SD**	**n**	**Min**	**Max**	**Mean ± SD**	**F_1,154_**	** *p* **
**TOTAL**	SL (mm)	83	4.28	9.01	6.04 ± 1.10	74	4.1	9.87	6.23 ± 1.43	5.42	*
	DW (mg)		0.1	2.89	0.49 ± 0.45		0.07	2.51	0.70 ± 0.61	26.78	**
	RADIUS (μm)		16.7	95.4	30.5 ± 13.50		13.8	89.3	33.00 ± 18.40	3.24	NS
	MIW (μm)		1.28	4.15	2.03 ± 0.60		1.06	4.64	2.16 ± 0.83	6.01	*
		**n**	**Min**	**Max**	**Mean ± SD**	**n**	**Min**	**Max**	**Mean ± SD**	**F_1,82_**	** *p* **
**PRE**	SL (mm)	49	4.28	7.06	5.34 ± 0.68	36	4.1	6.36	5.06 ± 0.57	0.04	NS
	DW (mg)		0.1	0.52	0.27 ± 0.12		0.07	0.64	0.30 ± 0.14	6.33	*
	RADIUS (μm)		16.7	33.4	23.26 ± 4.08		13.8	28	20.10 ± 3.61	5.66	*
	MIW (μm)		1.28	2.22	1.70 ± 0.25		1.06	2.06	1.53 ± 0.27	3.87	NS
		**n**	**Min**	**Max**	**Mean ± SD**	**n**	**Min**	**Max**	**Mean ± SD**	**F_1,69_**	** *p* **
**POST**	SL (mm)	34	6.08	9.01	7.04 ± 0.75	38	5.83	9.87	7.39 ± 1.01	11.02	**
	DW (mg)		0.21	2.89	0.81 ± 0.56		0.37	2.51	1.08 ± 0.63	20.9	**
	RADIUS (μm)		27.4	95.4	40.93 ± 15.36		23.5	89.3	44.85 ± 18.38	7.91	**
	MIW (μm)		1.78	4.15	2.51 ± 0.63		1.76	4.64	2.77 ± 0.73	6.52	**

**Table 4 animals-14-01319-t004:** Larval somatic data (SL, DW) and biometric otoliths (RADIUS, MIW) of intra-population (OPT, DEF) grouping by stage (PRE, POST, TOTAL) and years (GOM17, GOM18). ANCOVA analysis result (F, *p*) using AGE as covariate.** *p* < 0.01.

			OPT(+)				DEF(-)				ANCOVA		ANOVA	
**TOTAL**	GOM17		**n**	**Min**	**Max**	**Mean ± SD**	**n**	**Min**	**Max**	**Mean ± SD**	**F_1,52_**	** *p* **		
		SL (mm)	27	5.4	8.71	6.78 ± 0.78	28	4.3	9.01	5.52 ± 1.05	115.3	**		
		DW (mg)		0.3	2.13	0.69 ± 0.38		0.1	2.89	0.36 ± 0.54	140.8	**		
		RADIUS (μm)		23	79.3	36.30 ± 12.20		17.3	95.4	26.90 ± 15.20	63.98	**		
		MIW (μm)		1.8	4.14	2.41 ± 0.57		1.28	4.01	1.73 ± 0.54	34.14	**		
	GOM18		**n**	**Min**	**Max**	**Mean ± SD**	**n**	**Min**	**Max**	**Mean ± SD**	**F_1,59_**	** *p* **		
		SL (mm)	36	4.8	8.98	6.86 ± 1.03	26	4.1	9.87	5.77 ± 1.77	180.8	**		
		DW (mg)		0.29	2.42	0.85 ± 0.43		0.09	2.51	0.64 ± 0.83	132.4	**		
		RADIUS (μm)		15	77.5	36.90 ± 13.50		15.8	89.3	32.00 ± 25.20	79.73	**		
		MIW (μm)		1.3	4.64	2.51 ± 0.69		1.06	4.22	1.89 ± 0.96	40.56	**		
**PRE**			**n**	**Min**	**Max**	**Mean ± SD**	**n**	**Min**	**Max**	**Mean ± SD**	**F_1,27_**	** *p* **		
	GOM17	SL (mm)	15	4.3	7.06	5.79 ± 0.78	15	4.3	5.6	5.05 ± 0.36	55.85	**		
		DW (mg)		0.2	0.52	0.37 ± 0.11		0.1	0.25	0.18 ± 0.05	77.96	**		
		RADIUS (μm)		17	33.4	25.70 ± 4.25		17.9	26.8	21.78 ± 2.59	29.5	**		
		MIW (μm)		1.52	2.22	1.91 ± 0.18		1.28	1.83	1.53 ± 0.17	37.62	**		
			**n**	**Min**	**Max**	**Mean ± SD**	**n**	**Min**	**Max**	**Mean ± SD**	**F_1,23_**	** *p* **	**F_1,23_**	** *p* **
	GOM18	SL (mm)	11	4.8	6.2	5.59 ± 0.42	14	4.1	5.04	4.66 ± 0.27	152.18	**		
		DW (mg)		0.3	0.64	0.44 ± 0.12		0.09	0.38	0.20 ± 0.08			34.08	**
		RADIUS (μm)		15	26.4	22.74 ± 3.32		15.8	22.4	18.29 ± 2.29	54.73	**		
		MIW (μm)		1.26	2.06	1.79 ± 0.22		1.06	1.64	1.32 ± 0.17	51.11	**		
**POST**	GOM17		**n**	**Min**	**Max**	**Mean ± SD**	**n**	**Min**	**Max**	**Mean ± SD**	**F_1,26_**	** *p* **		
		SL (mm)	14	6.6	8.71	7.42 ± 0.68	11	6.08	9.01	6.78 ± 0.87	45.46	**		
		DW (mg)		0.5	2.13	0.97 ± 0.50		0.21	2.89	0.70 ± 0.77	45.38	**		
		RADIUS (μm)		30	79.3	45.90 ± 15.50		27.4	95.4	39.10 ± 19.50	41.71	**		
		MIW (μm)		2	4.14	2.84 ± 0.70		1.78	4.01	2.24 ± 0.63	26.98	**		
	GOM18		**n**	**Min**	**Max**	**Mean ± SD**	**n**	**Min**	**Max**	**Mean ± SD**	**F_1,26_**	** *p* **		
		SL (mm)	15	6.4	9.19	7.77 ± 0.79	14	5.83	9.25	6.87 ± 1.12	105.9	**		
		DW (mg)		0.5	2.51	1.22 ± 0.59		0.37	2.5	0.86 ± 0.71	56.48	**		
		RADIUS (μm)		28	85.5	49.10 ± 15.90		23.4	89.3	39.80 ± 22.20	57.97	**		
		MIW (μm)		2.2	4.64	3.16 ± 0.65		1.76	4.07	2.35 ± 0.73	39.15	**		

**Table 5 animals-14-01319-t005:** Inter-population results of Mann–Whitney U test (mean ± SE, number of larvae, Z adjusted and *p*) between years (GOM17 and GOM18) for maternal isotopic signatures (δ^15^N_maternal_, δ^13^C_maternal_). NS = Non-significant.

	GOM17		GOM18		MW—U Test	
	n	Mean ± SD	n	Mean ± SD	*Z-adjusted*	*p*
δ^15^N_maternal (estimated)_	49	12.30 ± 0.61	36	12.10 ± 0.72	1.66	NS
δ^13^C_maternal (estimated)_	49	−17.90 ± 0.38	36	−17.90 ± 0.28	0.07	NS

**Table 6 animals-14-01319-t006:** Intra-population (OPT vs. DEF) results of Mann–Whitney U test (mean ± SE, number of larvae, Z adjusted and *p*) between surveys (GOM17 and GOM18) for maternal isotopic signatures (δ^15^N_maternal_, δ^13^C_maternal_) estimated by equations based on samples of each survey sample. NS = Non-significant. ** *p* < 0.01.

	GOM17						GOM18					
	OPT +		DEF –		MW—U Test		OPT +		DEF –		MW—U Test	
	Mean ± SD	n	Mean ± SD	n	*Z-adjusted*	*p*	Mean ± SD	n	Mean ± SD	n	*Z-adjusted*	*p*
δ^15^N_maternal_	11.90 ± 0.44	15	12.50 ± 0.61	15	2.76	**	11.50 ± 0.33	11	12.60 ± 0.67	14	3.5	**
_(estimated)_												
δ^13^C_maternal_	−17.80 ± 0.22	15	−18.10 ± 0.46	15	−1.68	NS	−17.80 ± 0.35	11	−17.90 ± 0.17	14	−0.6	NS
_(estimated)_												

## Data Availability

Data supporting the results stated above can be sent to anyone requesting them from the authors.

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
