# Peer review of "Maternal Effects and Trophodynamics Drive Interannual Larval Growth Variability of Atlantic Bluefin Tuna (Thunnus thynnus) from the Gulf of Mexico"

_animals, 2024, doi:10.3390/ani14091319_

Round 1
Reviewer 1 Report
Comments and Suggestions for Authors
The manuscript is well written and structured and the research methodology is sound. However, there are areas that could benefit from improvement to enhance the clarity, rigour and overall impact of the study. Below are my specific comments and suggestions for revision:
1. The third paragraph of the Introduction provides a detailed introduction to why the Gulf of Mexico is an ideal habitat for ABFT. However, this content may not be closely related to the focal points of the present study. Consider condensing this section to more prominently highlight the key emphases within the Introduction.
2. The study's experimental design is robust. However, providing more information on the sample size and the representativeness of the samples collected would be helpful. Clarification on how the sample size was determined and whether it is sufficient to draw general conclusions about ABFT larval growth variability is needed.
3. The Methods section is comprehensive, but further details on the experimental procedures and measurement techniques would enhance reproducibility. Additional information on the protocols followed and equipment utilized is recommended.
4. Figures and tables are essential for understanding the results. Including additional statistical graphs to illustrate the variability in growth rates and isotopic signatures more effectively would be beneficial.
5. The Discussion section offers valuable insights into the study's findings. It could be further strengthened by explicitly articulating the study's implications for fisheries management and conservation efforts. Expand on the potential applications of your research and its contributions to ABFT ecology.
6. A more in-depth exploration of the study's limitations, including the potential impact of unmeasured environmental variables or genetic factors on larval growth rates, is needed to provide a more balanced perspective. This may include providing recommendations for future research to address these gaps.
7. The Conclusion should effectively summarize the main findings and their significance. Ensure that the final section of the manuscript encapsulates the research's contributions and suggests directions for future research.
8. While the manuscript is generally well-written, a thorough proofreading of the entire manuscript is recommended to correct any linguistic and grammatical inconsistencies and improve overall readability.
Reviewer 2 Report
Comments and Suggestions for Authors
The authors studied larvae of Atlantic bluefin tuna (Thunnus thynnus) from the Gulf of Mexico to evaluate the influence of maternal effects and trophodynamic factors on variability in larval growth.
The study compares two groups of larvae originating from the same general location over two years but exposed to different trophic conditions (richer in nutrients or more oligotrophic).
Materials were collected in 2017 and 2018 at 9 and 19 stations, respectively. Sampling methods are well described. In addition to larvae, zooplankton was sampled from each net for biomass determination, stable isotope analyses (SIA), and community analysis.
Temperature and water salinity were measured. Otoliths were removed to estimate the age of the larval. SIA analyses [N (δ15N) and C (δ13C)] of larvae and zooplankton were performed. Complete statistical analysis was carried out. Environmental variables, daily larval growth, otolith morphometry and stable isotope content have been analyzed.
The authors analyzed all parameters considering the total sample (TOTAL), as well as two subgroups - preflexion (PRE) and postflexion (POST) larvae, in order to distinguish between maternal factors and trophodynamic influences.
The results are well presented in tables and graphs, they seem objective and reasonable.
The discussion is quite detailed. A large number of references are provided (112).
The work can be accepted with minor changes.
The comments are as follows.
1. In Fig. 1, it is advisable to indicate the main currents and gyres, since they are mentioned in the text.
2. It is my understanding that there is one population of tuna in the Gulf of Mexico that has been studied for two years. In 2018 it was the same population, but a different year class or cohort. The use of "two populations", when comparing two consecutive years, is confusing. The terminology should be clarified.
3. Figure A1. It seems that the top of this curve should be moved to the left since 4.5 mm bar has the largest value (30%).
Round 2
Reviewer 1 Report
Comments and Suggestions for Authors
It is ok for acceptance.